# Global coordination of metabolic pathways in *Escherichia coli* by active and passive regulation

Karl Kochanowski[1,2,*] (ID), Hiroyuki Okano[3], Vadim Patsalo[4] (ID), James Williamson[4], Uwe Sauer[1] (ID) & Terence Hwa[3,5,**]

## Abstract

Microorganisms adjust metabolic activity to cope with diverse environments. While many studies have provided insights into how individual pathways are regulated, the mechanisms that give rise to coordinated metabolic responses are poorly understood. Here, we identify the regulatory mechanisms that coordinate catabolism and anabolism in *Escherichia coli*. Integrating protein, metabolite, and flux changes in genetically implemented catabolic or anabolic limitations, we show that combined global and local mechanisms coordinate the response to metabolic limitations. To allocate proteomic resources between catabolism and anabolism, *E. coli* uses a simple global gene regulatory program. Surprisingly, this program is largely implemented by a single transcription factor, Crp, which directly activates the expression of catabolic enzymes and indirectly reduces the expression of anabolic enzymes by passively sequestering cellular resources needed for their synthesis. However, metabolic fluxes are not controlled by this regulatory program alone; instead, fluxes are adjusted mostly through passive changes in the local metabolite concentrations. These mechanisms constitute a simple but effective global regulatory program that coarsely partitions resources between different parts of metabolism while ensuring robust coordination of individual metabolic reactions.

**Keywords** [13]C flux analysis; Crp; metabolomics; proteomics; regulation analysis

**Subject Categories** Metabolism; Microbiology, Virology & Host Pathogen Interaction

**Mol Syst Biol. (2021) 17: e10064**

## Introduction

In nature, microbes often encounter unpredictable changes in nutrient availability. To sustain growth when facing such environmental variations, microbes need to coordinate their metabolic supply of biomass precursors, energy, and redox factors (Chubukov *et al,* 2014) and coordinate the allocation of their proteome for metabolism and macromolecular synthesis (Scott *et al,* 2010; Erickson *et al,* 2017). In principle, microbes can mobilize a vast arsenal of regulatory mechanisms including transcriptional control (Kochanowski *et al,* 2013a), covalent posttranslational modifications (Oliveira & Sauer, 2012; Pisithkul *et al,* 2015; Su *et al,* 2016), and non-covalent binding by small molecules to proteins (Li *et al,* 2013; Kochanowski *et al,* 2015), and many recent case studies have elucidated the role of these mechanisms in regulating individual metabolic pathways (Kao *et al,* 2004; Semsey *et al,* 2007; Yuan *et al,* 2009; Li *et al,* 2010; Kotte *et al,* 2010; Madar *et al,* 2011; Grüning *et al,* 2011; Cho *et al,* 2012; Oliveira & Sauer, 2012; Reaves *et al,* 2013; van Heerden *et al,* 2014; Gerosa *et al,* 2015; Kim *et al,* 2018; Sander *et al,* 2019; Okano *et al,* 2020). Mounting an appropriate response, however, requires not only regulation of a single pathway but coordinated regulation of a larger network (Wayman & Varner, 2013; Chubukov *et al,* 2014). Examples of such coordination are the bacterial stringent response to nutrient starvation that triggers the general activation of stress genes (Chatterji & Kumar Ojha, 2001; Potrykus & Cashel, 2008), the Crp-dependent activation of catabolic genes in carbon limitation (You *et al,* 2013), and the coordination of carbon and nitrogen assimilation by 2-oxoglutarate (Doucette *et al,* 2011; Kim *et al,* 2012). However, while these and other (Cho *et al,* 2012; Federowicz *et al,* 2014; Goel *et al,* 2015; Olin-Sandoval *et al,* 2019) examples unravel some of the more global coordination mechanisms, it remains unclear how a coordinated metabolic response emerges mechanistically from the interplay of the cell's various regulatory circuits (Chubukov *et al,* 2014).

Here we aim to identify the regulatory mechanisms that enable *Escherichia coli* to coordinate metabolic activity in response to different modes of nutrient limitation. Specifically, we focus on two

1 Institute of Molecular Systems Biology, ETH Zurich, Zurich, Switzerland
2 Life Science Zurich PhD Program on Systems Biology, Zurich, Switzerland
3 Department of Physics, University of California at San Diego, La Jolla, CA, USA
4 Department of Integrative Structural and Computational Biology, and The Skaggs Institute for Chemical Biology, The Scripps Research Institute, La Jolla, CA, USA
5 Institute for Theoretical Science, ETH Zurich, Zurich, Switzerland
*Corresponding author. Tel: +34 693 015 290; E-mail: karl.kochanowski@gmail.com
**Corresponding author. Tel: +1 858 534 7263; E-mail: hwa@ucsd.edu

orthogonal challenges, namely limitation of carbon influx ("catabolic limitation") and limitation in the ability to synthesize all amino acids while carbon is in excess ("anabolic limitation"). Previously, we constructed strains with titratable control of glucose uptake to implement catabolic limitation and strains with titratable control of glutamate synthesis (and hence limiting the transamination flux needed for the synthesis of all amino acids, Appendix Fig S1) to implement anabolic limitation. Importantly, these limitations can be conveniently imposed to mimic carbon- or ammonium-limited continuous cultures (Goel *et al*, 2015; Hackett *et al*, 2016) while maintaining identical batch-like experimental conditions, using medium with saturating amounts of glucose and ammonium (You *et al*, 2013; Hui *et al*, 2015). It was found that *E. coli*'s regulatory response to these limitations leads to a striking global proteome re-allocation (Hui *et al*, 2015): Catabolic and anabolic limitations are accompanied by a general increase and decrease in abundance of catabolic proteins, respectively, a pattern that is reversed for anabolic proteins.

How is this coordinated gene regulatory response established mechanistically? It is clear that transcriptional regulation of catabolic proteins in *E. coli* is largely driven by the transcriptional activator Crp (You *et al*, 2013; Gerosa *et al*, 2015). When carbon limits growth, the accumulation of cyclic AMP, the small-molecule activator of Crp, causes the general increase in the expression of catabolic proteins. Conversely, in anabolic limitation—when the availability of external carbon exceeds the cell's ability to synthesize biomass precursors such as amino acids—the accumulation of alpha-keto acids, such as 2-oxoglutarate (Doucette *et al*, 2011; You *et al*, 2013), inhibits the production of cyclic AMP (cAMP), thereby reducing the expression of catabolic proteins (Fig 1A). While this Crp-driven regulatory circuit provides a plausible explanation for the transcriptional response of catabolic proteins, it is unclear how anabolic gene expression is regulated (Fig 1A). Given the reversed behavior of catabolic and anabolic proteins (Hui *et al*, 2015), a parsimonious mechanism would have Crp (which activates catabolic proteins) also serving as a repressor of anabolic proteins. However, there are only few reported instances of anabolic proteins being under Crp control (Shimada *et al*, 2011; Santos-Zavaleta *et al*, 2019).

How else, then, could the proteome response of catabolism and anabolism be coordinated? As we will show, this coordination is achieved by a passive mode of regulation which originates from the inherent competition for limiting cellular resources (Scott *et al*, 2010) and which depends on Crp directly activating catabolism and indirectly repressing anabolism. Moreover, by integrating this gene regulatory program with the large-scale quantification of metabolic fluxes and metabolite concentrations, we demonstrate that passive local adjustments in enzyme saturation play a pivotal role in implementing a coordinated metabolic response for individual metabolic reactions.

# Results

### A single transcription factor coordinates the global transcriptional response to nutrient limitation

To understand how the coordinated proteome response of catabolism and anabolism to nutrient limitation could be established mechanistically, we started from the observation that genes compete

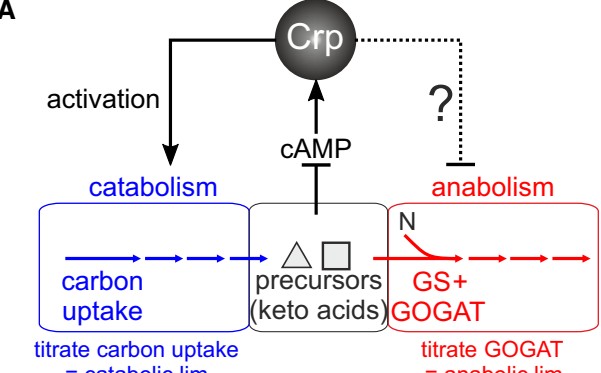

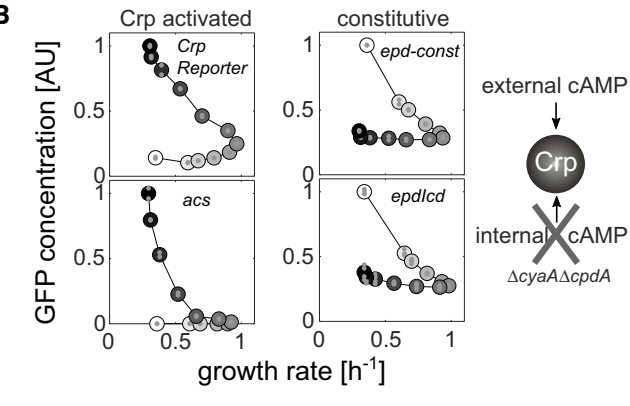

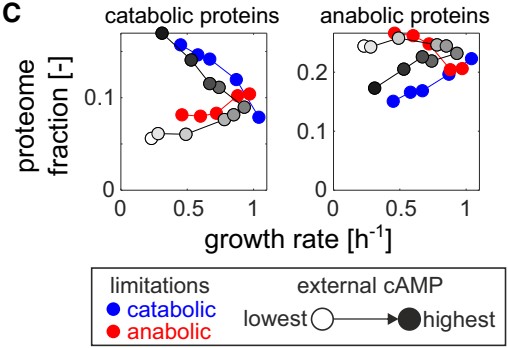

**Figure 1.  Transcriptional coordination of catabolism and anabolism.**

A  Schematic of metabolism—separated into catabolism (blue), central metabolic precursors (e.g., alpha-keto acids, black), and anabolism (red)—and the cyclic AMP (cAMP)-dependent Crp regulatory circuit in *Escherichia coli*. By titrating the expression of carbon uptake proteins, catabolic limitation can be imposed externally. Conversely, by titrating the expression of the enzyme glutamate synthase (GOGAT), a limitation of anabolic capacity can be imposed externally. N: nitrogen. GS: glutamine synthase. See Appendix Fig S1 and Appendix Text 1 for detailed description of titration strains.

B  Steady-state expression of Crp-activated (left panel) or constitutive (right panel) fluorescent transcriptional reporters at varying external cyclic AMP concentrations (white and black colors denote lowest and highest external cyclic AMP concentration, respectively) in a strain that cannot produce or degrade endogenous cAMP (NQ1399, ΔcyaAΔcpdA). Small gray circles: individual biological replicates (*n* = 2–4). Large circles: mean across replicates.

C  Catabolic and anabolic fraction of *E. coli*'s proteome in catabolic (blue) and anabolic (red) limitation (data from Ref. Hui *et al*, 2015 with few excluded proteins, see Appendix Text 3), as well as at varying external cAMP concentrations in strain NQ1399.

with each other for the limited capacity of the cellular protein expression machineries (Maaløe, 1979; Vind *et al*, 1993; Bremer & Dennis, 1996; Scott *et al*, 2010; Gerosa *et al*, 2013; Kafri *et al*, 2016; Borkowski *et al*, 2016). For example, expression of unneeded proteins was found to reduce the translation of other proteins, presumably due to competition for ribosomes (Vind *et al*, 1993; Scott *et al*, 2010). Based on these studies, we hypothesized that activation of hundreds of genes in the Crp regulon (Santos-Zavaleta *et al*, 2019), which constitutes up to ~30% of the cellular proteome (Hui *et al*, 2015; Schmidt *et al*, 2016), would sequester a large fraction of the cell's expression machinery capacity, thereby indirectly reducing the expression of non-Crp targets during catabolic limitation. Conversely, in anabolic limitation, where Crp is inactivated, other genes or transcripts would have access to a greater share of the expression machinery. This hypothesis is consistent with the negative and positive correlation of the intracellular cAMP concentrations with the growth rate for the catabolic- and anabolic-limited cultures, respectively (Appendix Fig S2).

To test whether Crp could indirectly repress the expression of non-Crp target proteins, and to mimic the divergent response of Crp in the two limitations, we constructed a strain (NQ1399) lacking endogenous cAMP production, such that Crp activity can be controlled externally through supplementation of cAMP (Kuhlman *et al*, 2007; Towbin *et al*, 2017) (Appendix Text S2). Using this "cAMP titration" strain, we quantified the activity of two synthetic constitutive promoters without any known transcription factor binding site (Gerosa *et al*, 2013; Kochanowski *et al*, 2017) and two exclusively Crp-activated promoters at varying external cAMP concentrations using fluorescent reporter plasmids (Gerosa *et al*, 2013; Kochanowski *et al*, 2017). As reported previously (Towbin *et al*, 2017), titration of cAMP to cultures of this strain caused a monotonic increase in the activity of Crp-activated promoters (Appendix Fig S3) and also reduced the growth rate for low as well as high cAMP concentrations (Appendix Fig S4). In agreement with our hypothesis, this pattern was reversed in strains with constitutive promoters, whose activity was lowest at the highest cAMP concentration (Appendix Fig S3). When plotting the activity of these promoters against the growth rate, we observed reversed growth-dependent patterns for catabolic and constitutive promoters (Fig 1 B). These reversed expression patterns were also found in (Crp-activated) catabolic tricarboxylic acid (TCA) cycle compared to amino acid biosynthesis and constitutive-like glycolytic promoters (Appendix Figs S5 and S6). Thus, these fluorescent reporter data showed that activation of Crp can reduce the expression of non-Crp targets. Moreover, the similarity of non-Crp target expression to that of synthetic constitutive promoters suggests that the effect of Crp on non-Crp targets is indeed indirect.

To test whether such changes in Crp activity would also be sufficient to modulate the anabolic proteome fraction, we quantified the proteome response of the cAMP titration strain using shotgun proteomics. First, we tested whether deletion of adenylate cyclase (encoded by *cyaA*) and phosphodiesterase (encoded by *cpdA*) employed in the cAMP titration strain (see Appendix Table S1) affected the proteome independently of their effect on cAMP production. For this purpose, we grew the cAMP titration strain in glucose medium at an external cAMP concentration that matches the wild-type growth rate (Appendix Fig S4). The vast majority (88%) of proteins were within twofold of the wild-type

concentrations, confirming that deletion of *cyaA* and *cpdA* has a limited impact on the proteome independently of cAMP (see Appendix Text 3 and Appendix Fig S7A and B). The few exceptions included proteins belonging to flagella and chemotaxis, which notably are under the control of promoters that frequently acquire mutations during strain development (Parker *et al*, 2019), and proteins whose operons are located close to the deletion loci (i.e., IlvB and RbsB), which could be attributed to genetic differences in the donor strain used to generate the gene deletions (Lyons *et al*, 2011) (see Appendix Text 2). Second, we titrated the cAMP concentration below and above the level required to yield wild-type-like growth rate. This titration affected the abundance of hundreds of proteins (Appendix Fig S7C) and led to a reversed global response of catabolic and anabolic proteins (gray points in Fig 1C, Appendix Fig S7D). Gratuitous Crp activity at high cAMP concentrations (above wild-type level) caused an overall increase in the abundance of catabolic proteins with a concomitant reduction of anabolic protein abundance, a pattern which was reversed with diminished Crp activity at low cAMP concentrations (below wild-type level).

Quite strikingly, the proteome response of the cAMP titration data showed good agreement with that of catabolic/anabolic limitation at the same growth rate, respectively; compare gray symbols with blue/red symbols in Fig 1C. In particular, catabolic proteins (Fig 1C, left) responded similarly when comparing the limitations and cAMP titration. This overlap was less pronounced for anabolic proteins (Fig 1C, right), in particular when comparing carbon limitation and gratuitous cAMP titration (blue circles and dark gray circles, see also Appendix Fig S8 for examples of individual proteins), suggesting that additional pathway-specific transcriptional regulators may further modulate the indirect regulatory effect by Crp. Notably, not all proteins were affected by the titration of Crp activity: Ribosomal proteins maintained their strict relationship with the growth rate (Appendix Figs S9 and S10), presumably due to additional compensatory regulation by ppGpp and DksA (Paul *et al*, 2004; Lemke *et al*, 2011; Bosdriesz *et al*, 2015). In addition, the sulfate assimilation proteins CysI and CysJ, and the methionine forming enzyme homocysteine transmethylase (MetE), showed a positive correlation with growth rate across all tested conditions (Appendix Fig S10). Since MetE, with its very low catalytic activity and high abundance, makes methionine biosynthesis a highly resource-demanding process (Li *et al*, 2014), a plausible explanation is that additional dedicated regulation, mediated presumably by the methionine biosynthesis regulators MetR and MetJ, further adjusts MetE expression to match the growth-dependent demand for methionine.

Our findings thus suggest a passive mode of regulating protein expression. We propose that Crp, the main transcriptional regulator of catabolic proteins, indirectly regulates the expression of anabolic proteins and thereby coordinates the expression of catabolism and anabolism. While the data presented here do not prove unequivocally that Crp regulation is the direct cause of the reversed patterns of catabolic and anabolic proteins in response to nutrient limitation, the good agreement of the proteome response to nutrient limitation and Crp activity titration, together with the response pattern of the constitutively expressed proteins, suggests that this model of regulation provides the most parsimonious mechanism consistent with all available data.

**Global transcriptional program alone cannot explain the observed metabolic flux**

The above passive mode of gene regulation suggests a potential mechanism for the global coordination of catabolism and anabolism. Next, we investigate whether this regulatory program is sufficient to account for *E. coli*'s metabolic flux requirement for growth under the imposed nutrient limitations. Toward this end, we cultivated the aforementioned titratable strains (Appendix Fig S1) at different induction levels (8 induction levels per limitation) to impose different degrees of catabolic or anabolic limitation while keeping the media composition constant. To obtain a comprehensive picture of the steady-state flux response at each induction level, we determined extracellular exchange rates of over 20 metabolites and ten ratios of intracellular fluxes in central metabolism by $^{13}$C flux analysis (Zamboni *et al*, 2009) (Appendix Figs S11 and S12). Using these data and the growth requirements as constraints for flux balance analysis (Schellenberger *et al*, 2011; Heirendt *et al*, 2019), we obtained markedly different flux estimates between the catabolic and anabolic limitations (Appendix Fig S13A). In particular, there was a general shift in fermentation strategy between catabolic and anabolic limitations (Fig 2A): Consistent with many previous studies such as in carbon-limited chemostats (Varma & Palsson, 1994; Nanchen *et al*, 2006; Valgepea *et al*, 2010; Renilla *et al*, 2012; Basan *et al*, 2015), increasing growth rates under catabolic limitation caused a shift from full respiration to acetate overflow metabolism. Under anabolic limitation, however, acetate secretion occurred even at the lowest tested growth rate, as reported before in nitrogen-limited chemostat cultures (Sauer *et al*, 1999), but unexpectedly we also observed substantial 2-oxoglutarate secretion. To our knowledge, 2-oxoglutarate secretion has never been reported in nitrogen-limited chemostats. A possible contributor to this secretion may be a difference in the promoter region of the 2-oxoglutarate transporter KgtP in our strain compared to the common laboratory strain MG1655 (Lyons *et al*, 2011), although recent studies have demonstrated that many metabolites other than acetate can be secreted by *E. coli* (Paczia *et al*, 2012; Reaves *et al*, 2013), notably including 2-oxoglutarate (Paczia *et al*, 2012). Also, with up to 50% of the consumed glucose secreted as acetate and 2-oxoglutarate secretion under anabolic limitation (Appendix Fig S12), the resulting reduction in biomass yield we obtained was consistent with that observed in nitrogen-limited chemostat cultures (Sauer *et al*, 1999). The secretion of all other metabolites was at least an order of magnitude lower. As a result, most fluxes (in particular biosynthetic fluxes outside of central carbon metabolism) were found to scale with the growth rate regardless of the imposed limitation (Appendix Fig S13B and C). Moreover, most FBA-estimated fluxes were well constrained by the measured physiology as determined by flux variability analysis (Appendix Fig S13D).

With the quantified metabolic fluxes at hand, we asked whether the observed flux changes could be explained by corresponding protein concentration changes. For this purpose, we used the previously established theoretical framework of regulation analysis that quantifies the contribution of individual regulatory layers to the observed flux changes (Rossell *et al*, 2006; Chubukov *et al*, 2013; Gerosa *et al*, 2015) (see Appendix Text 3 for detailed description). For example, the contribution of gene expression can be quantified as the log–log slope between flux and protein concentration changes

(Fig 2B), and is referred to as the protein regulation coefficient, $\rho_P$. $\rho_P = 1$ signifies that the observed flux changes can be fully explained by changes in protein concentration. Conversely, $\rho_P < 0$ signifies that flux and protein concentration change in opposite directions. For the catabolic and anabolic limitations, we determined protein regulation coefficients for 202 unique reactions by linear regression for all reactions that carry flux under the tested conditions and where at least one of the associated proteins was quantified. A shift toward negative protein regulation coefficients was apparent for the anabolic limitation compared to catabolic limitation, signifying that protein and flux change in opposite directions (Fig 2C). This shift was particularly pronounced in biosynthetic reactions, but occurred also in central reactions (Fig 2D, top row). Only 18 and 7% of all reactions had protein regulation coefficients around one under carbon and anabolic limitation, respectively. While systematic errors in proteomic analysis for weakly expressed proteins may play some role in these results, overall, the data show that observed flux changes were rarely accompanied by matching changes in protein concentrations. Among the few notable exceptions (Fig 2D, bottom row) was the aforementioned methionine biosynthesis enzyme MetE, suggesting that *E. coli* adjusts the expression of MetE to minimize the cost of methionine biosynthesis, in particular under catabolic limitation (Li *et al*, 2014).

The poor agreement between flux and proteome response observed here is in line with previous observations showing that gene expression is typically a poor predictor of flux changes (Chubukov *et al*, 2013; Valgepea *et al*, 2013; Kochanowski *et al*, 2013a; Gerosa *et al*, 2015; Goel *et al*, 2015; O'Brien *et al*, 2016; Hackett *et al*, 2016), and suggests that the gene regulatory program characterized above is rarely sufficient to regulate catabolic and anabolic metabolic fluxes. In particular, under anabolic limitation we observed that fluxes and proteins tended to change in opposite directions for many biosynthetic reactions (i.e., decrease in biosynthetic flux that is accompanied by an increase in protein concentration).

**Passive regulation of enzyme activity through altering enzyme saturation**

If the gene regulatory program alone is not sufficient to explain the observed metabolic fluxes, which other mechanisms could? One way how enzyme activity could be adjusted is by altering enzyme saturation through changes in the corresponding substrate concentrations. This "passive" mechanism of flux regulation provides a means to effectively buffer changes in enzyme concentration while keeping the flux constant (Fendt *et al*, 2010). Analogous to gene regulation, regulation analysis can quantify the effect of enzyme saturation on flux as the saturation regulation coefficient $\rho_S$, based on a power-law approximation of the non-linear relationship between substrate concentration and reaction rate (Chubukov *et al*, 2013; Gerosa *et al*, 2015) (see Appendix Text 3).

To test whether altered enzyme saturation could account for the observed flux responses, we quantified the relative intracellular concentration of 430 unique metabolites by untargeted metabolomics (Fuhrer *et al*, 2011) and obtained consistent results from targeted quantification of 40 central metabolites (Appendix Fig S14). Compared to the flux response, the metabolome showed more complex patterns that differed between the two limitations, as revealed by K-means clustering (Fig 3A). In some of these groups,

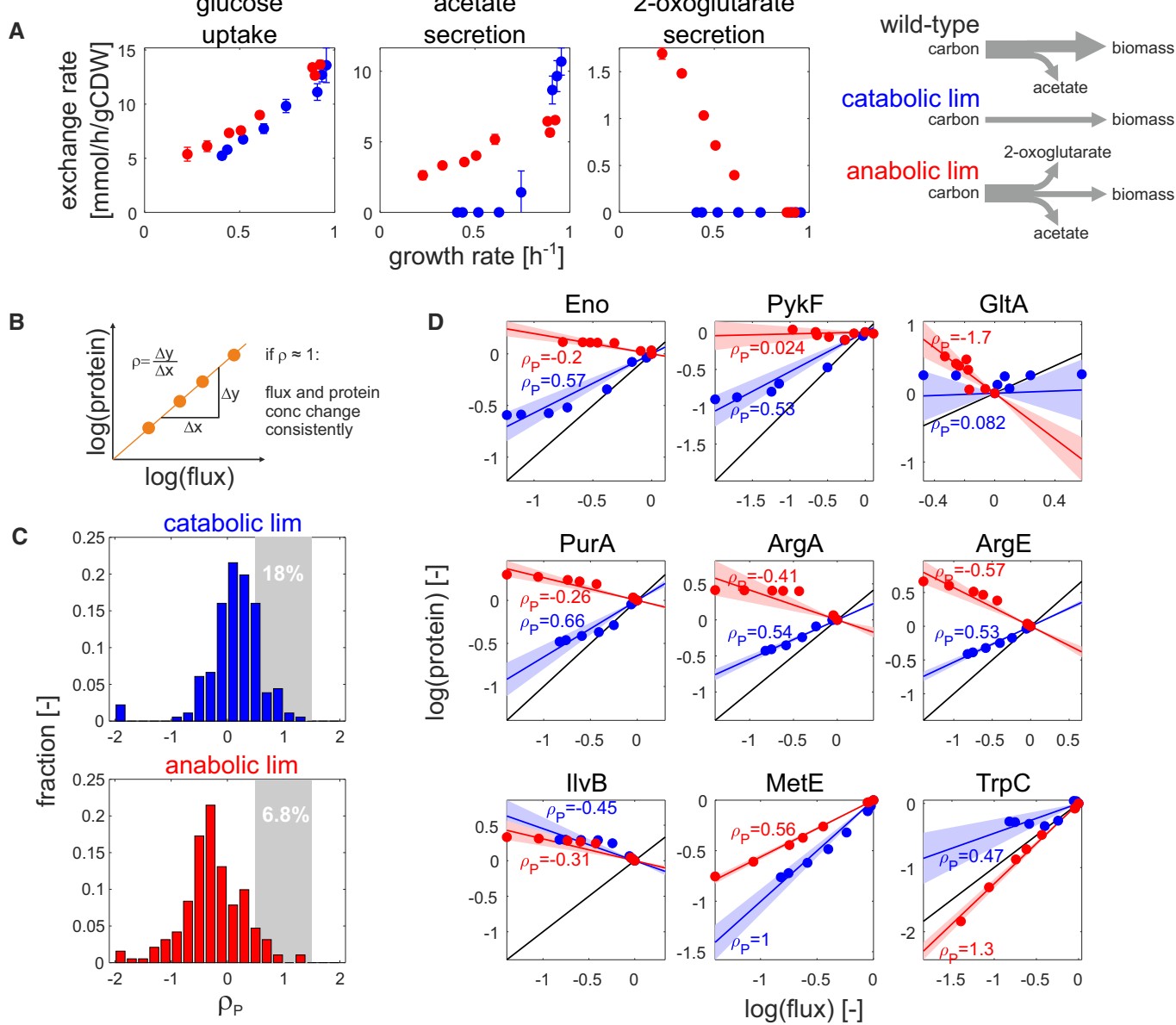

**Figure 2. Regulation analysis of flux–protein relationship in nutrient limitations.**

A Specific uptake and secretion rates of glucose (left), acetate (middle), and 2-oxoglutarate (right) in catabolic (blue) and anabolic (red) limitation. Data shown are the mean of three biological replicates, and error bars denote standard deviation.

B Schematic description of regulation analysis: The regulation coefficient ρ denotes the degree to which flux and protein concentration of a given reaction change consistently across conditions (ρ = 1: full proportionality of flux and protein changes). See Appendix Text 3 for detailed description.

C, D Distribution of protein regulation coefficients ($\rho_P$) for catabolic (upper panel) and anabolic (lower panel) limitation. Protein regulation coefficients were determined separately for each limitation by linear regression. For reactions that are associated to more than one protein, the final protein regulation coefficients were calculated as the average regulation coefficients across all measured proteins (total number of considered reactions: 202). The fraction of reactions showing consistent flux and protein changes (regulation coefficients between 0.5 and 1.5) is highlighted by gray areas with corresponding percentages. Individual examples are shown in (D). Blue and red lines denote the estimated regulation coefficient $\rho_P$ (calculated by linear regression) between measured flux and protein concentrations for catabolic and anabolic limitation, respectively. Shaded areas denote the standard error of each estimate. Black lines denote full proportionality. Top panel: central metabolic reactions. Middle panel: biosynthetic reactions showing a shift from positive to negative $\rho_P$ for carbon and anabolic limitation, respectively. Bottom panel: biosynthetic reactions which deviate from this behavior.

only one limitation caused large changes in metabolite concentrations, for example, the L-glutamine (cluster 7) and 2-oxoglutarate (cluster 1) groups accumulated only in anabolic limitation. Other metabolite groups, such as the one including fructose-1,6-bisphosphate (FBP, cluster 4), were affected by both limitations, albeit in some cases in opposite directions (e.g., clusters 6, 9, and 10).

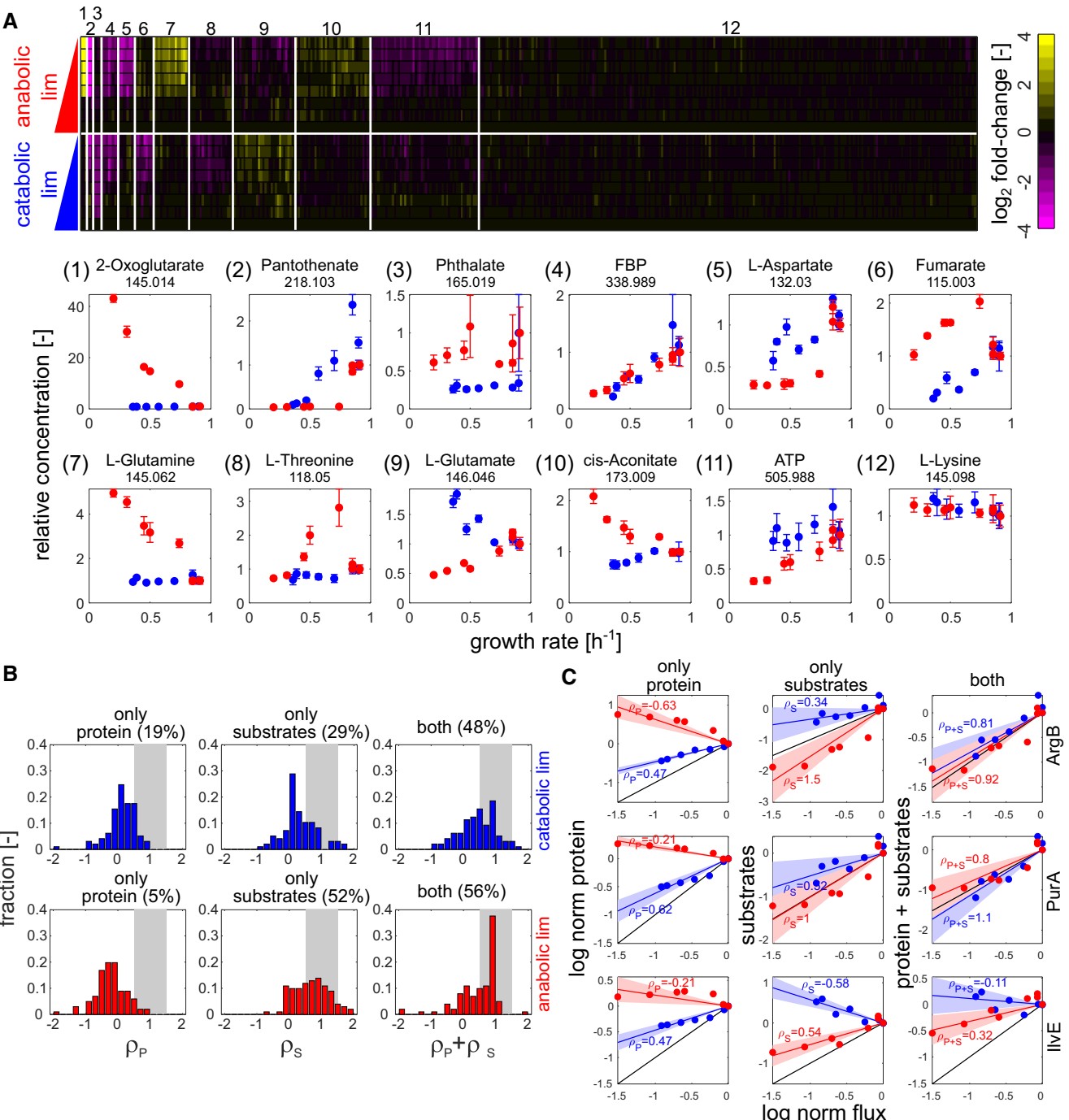

**Figure 3. Impact of enzyme saturation on flux regulation.**

A   Intracellular metabolome response (of 430 annotated ions) in mid-exponential growth as determined by FIA-TOF mass spectrometry. Upper panel: heatmap of $\log_2$ fold changes (relative to NCM3722 wild type) for both limitations. Annotated ions were sorted by K-means clustering of $\log_2$ fold changes, using squared Euclidian distance as metric (Arthur & Vassilvitskii, 2007). Annotated ions showing little variation across conditions (coefficient of variance across all conditions < 30%) were filtered out before clustering and assigned to cluster 12. Lower panel: exemplary ions for each cluster plotted (in linear scale) against the respective growth rate. Numbers underneath metabolite names denote m/z. Error bars denote standard deviation of three biological replicates.

B, C   Distribution of regulation coefficients quantifying the contribution of changes in protein concentration (left column) and enzyme saturation (middle column) to the explanation of observed flux changes, as well as their combined effect (right column). Upper panel: catabolic limitation. Lower panel: anabolic limitation. Only reactions for which at least one isoenzyme as well as all substrates had been quantified were considered in the analysis (total number of considered reactions: 108). The fraction of reactions with regulation coefficients between 0.5 and 1.5 is highlighted by gray areas. Individual exemplary reactions are shown in (C). Blue and red lines denote the estimated regulation between measured flux, protein, and metabolite concentrations for catabolic and anabolic limitation, respectively. Shaded areas denote the standard error of each corresponding estimate. Black lines denote full proportionality. ρ: corresponding regulation coefficients.

Nevertheless, more than 50% of the quantified metabolites were not significantly affected by either limitation, including most amino acids (cluster 12). Notably, the observed metabolome response to the genetically implemented limitations used here matches previous reports utilizing analogous environmental limitations, such as the consistent dramatic accumulation of 2-oxoglutarate amid largely constant amino acid concentrations found in nitrogen-limited cultures of the same *E. coli* background strain (Yuan *et al*, 2009).

Surprisingly, L-glutamate itself, whose production rate is reduced in the anabolic limitation used in this work (Appendix Fig S1), proved to be quite resilient against either limitation, consistent with previous findings in *Salmonella typhimurium* (Ikeda *et al*, 1996). Particularly in the anabolic limitation, which caused an up to five-fold growth rate reduction, its concentration was only twofold reduced and never dropped below 10 mM (Appendix Fig S14E), which is above the reported $K_M$ values for most glutamate-consuming reactions (Schomburg *et al*, 2002).

Based on this metabolome response, we quantified the saturation regulation coefficient $\rho_S$ for over 100 reactions (separately for each limitation), only considering those reactions for which *all* substrates were quantified. In both limitations, the vast majority of $\rho_S$ were positive, meaning that flux and substrate concentrations changed in the same direction (Fig 3B, middle column). Analysis of the combined effect of gene expression and enzyme saturation showed that the sum of $\rho_P$ and $\rho_S$ accounts for about half of all flux changes in both limitations (i.e., $\rho_P + \rho_S \approx 1$ for ~ 50% of the examined reactions, see Fig 3B, right column), indicating that the multiplicative effects of enzyme abundance and enzyme saturation (as in, e.g., Michaelis–Menten relation) account for these reactions. However, the relative contribution of enzyme level changes and enzyme saturation differed between the two limitations: In carbon limitation, both mechanisms had a similar impact, whereas in the anabolic limitation, the contribution of enzyme saturation was dominant. This difference in regulatory strategy was typically maintained within a metabolic pathway (see arginine biosynthesis pathway as an example, Appendix Fig S15).

Taken together, these data show that passive regulation through enzyme saturation is pivotal for coordinating enzyme activity across conditions. This finding is consistent with previous computational studies highlighting the importance of metabolic regulation (that is, regulation of enzyme activity by, e.g., enzyme saturation and allosteric regulation) for the coordination of *E. coli* metabolism (Millard *et al*, 2017).

## Discussion

In this study, we aimed to identify the mechanisms underlying *E. coli*'s coordinated response to different metabolic limitations. From the large-scale quantification of metabolic fluxes, metabolites, and proteins, in genetically implemented catabolic and anabolic limitations, we identify two mechanisms that facilitate the global as well as local coordination of metabolic activity (Fig 4). First, a global gene regulatory program coordinates the expression of catabolism and anabolism, largely through the activity of a single transcription factor, Crp. This regulatory program however does not exactly match protein levels to the required flux changes for many reactions. Second, the mismatches between protein and flux changes are

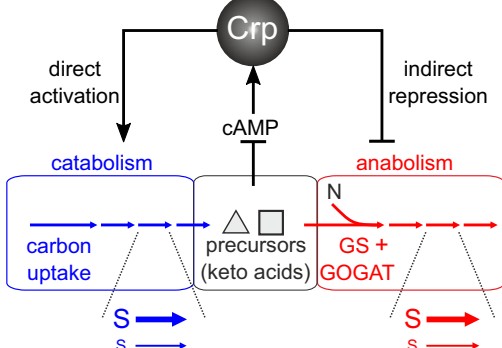

**Figure 4. Schematic summary of mechanisms implementing the global coordination of metabolism identified in this study.**

Coordination in the expression of catabolic and anabolic proteins is achieved by a single transcription factor, Crp, which directly induces catabolic proteins and indirectly represses anabolic proteins under catabolic limitation, through the competition for scarce expression machinery capacity (mechanism 1, see also Box 1). This approximate gene regulatory program is further adjusted locally through changes in metabolite concentrations, which alter each enzyme's saturation. At a given enzyme concentration, high substrate concentration (= high enzyme saturation, symbolized by large S) leads to a high reaction rate (symbolized by a thick horizontal arrow). Conversely, low substrate concentration (small S) reduces the reaction rate (thin horizontal arrow).

adjusted locally through passive changes in enzyme saturation. These findings provide insights on mechanistic implementations of global resource allocation predicted previously by the phenomenological theory of bacterial growth control (Appendix Text 4 and Appendix Figs S17–S19).

Our results suggest that Crp exerts its effect on catabolic and anabolic proteins through both direct and indirect regulation: Catabolic proteins are directly induced by Crp under catabolic limitation, while anabolic protein expression is repressed indirectly. Conversely, Crp induction of catabolic proteins is reduced under anabolic limitation (due to increased 2-oxoglutarate (You *et al*, 2013)), while anabolic protein expression is indirectly increased, exhibiting similar behaviors as the constitutive (i.e., unregulated) reporters. Importantly, the indirect mode of regulation reported here has broad implications beyond the coordination of catabolism and anabolism by Crp. A common approach to identify the targets of a transcriptional regulator is to identify genes whose expression changes upon its deletion (e.g., Wang *et al*, 2018). Our results suggest that deletion of transcriptional regulators, in particular global regulators, may also affect the expression of non-target genes, thus confounding the regulatory networks resulting from such efforts. Nevertheless, this indirect regulation of non-Crp targeted genes can be overridden by additional designated regulation, as exemplified by the regulation of the ribosomal proteome fraction, which maintains its strict dependency on growth rate in both of the imposed limitations (Appendix Fig S9). Future studies may use this work as a starting point to identify the regulatory mechanisms responsible for

overriding the indirect regulation by Crp in other proteome fractions, e.g., those which do not change expression across either catabolic or anabolic limitations (Hui *et al*, 2015).

How this indirect mode of regulation is achieved mechanistically is currently unclear. In principle, this mechanism could be the result of competition for limited capacity of expression machinery at the transcription or translation level (Box 1). The proposed models yield the same result at the proteome level, and identifying whether any of these—or alternative—models are responsible for this indirect mode of regulation is left for future studies.

Although gene expression and enzyme saturation accounted for about 50% of the observed flux changes in a simple multiplicative model for both limitations, many flux changes could not be explained simply by these two mechanisms alone. For example, most glutamate-dependent transamination reactions—in which the amination of a metabolite is coupled to the deamination of glutamate to 2-oxoglutarate (see Appendix Fig S1C)—were poorly

accounted for by changes in protein and substrate concentrations alone, resulting in regulation coefficients far from 1 (see IlvE as example in Fig 3C, bottom row). A possible explanation is that these reactions are affected by concerted changes in both substrate and product concentrations (Hackett *et al*, 2016). Although the lack of complete metabolite and proteome data for many reactions does not allow us to assess the impact of reaction products across a large number of reactions, our results already suggest that the glutamate-dependent transamination reactions show a massive shift in the ratio of 2-oxoglutarate (reaction product of most transamination reactions) to glutamate (reaction substrate) under anabolic limitation, causing a "thermodynamic choke-point" (Appendix Fig S16). In contrast, transamination reactions which use fumarate instead of 2-oxoglutarate as a product (and aspartate instead of glutamate as a substrate) are explained well by gene expression and enzyme saturation (see ArgG in Appendix Fig S15 as an example). Importantly, these findings also suggest a possible

---

**Box 1.  Two plausible mechanisms of indirect repression of non-Crp targets by the transcriptional activator Crp**

Model 1 (panel A), termed "limiting transcriptional capacity", is based on the premise that protein expression is limited by different promoters in the cell competing for the same pool of free RNA polymerase molecules for transcription initiation. In this case, activation of Crp, which exerts its activating effect by directly recruiting RNA polymerase to catabolic genes (Lawson *et al*, 2004), would lead to a reduction in free RNA polymerase to initiate the transcription of other genes not activated by Crp, consequently causing a reduction in the expression of these genes. As a result, in this model indirect repression of genes not activated by Crp is established mechanistically as a Crp-dependent re-arrangement of the cell's transcriptome. Model 2 (panel B), termed "limited translational capacity", assumes that translation (i.e., number of free ribosomes) is the rate-limiting step in protein expression. In this case, activation of Crp would lead to increase in the transcript abundance of Crp-activated genes without effecting the transcript abundance of genes not activated by Crp. If Crp and non-Crp target transcripts compete for limiting ribosomes, an increase in number of Crp target transcripts would effectively reduce the availability of free ribosomes to initiate the translation of non-Crp target transcripts. Thus, in this model Crp-activated transcripts effectively act as competitive inhibitors of the translation of non-Crp-activated transcripts.

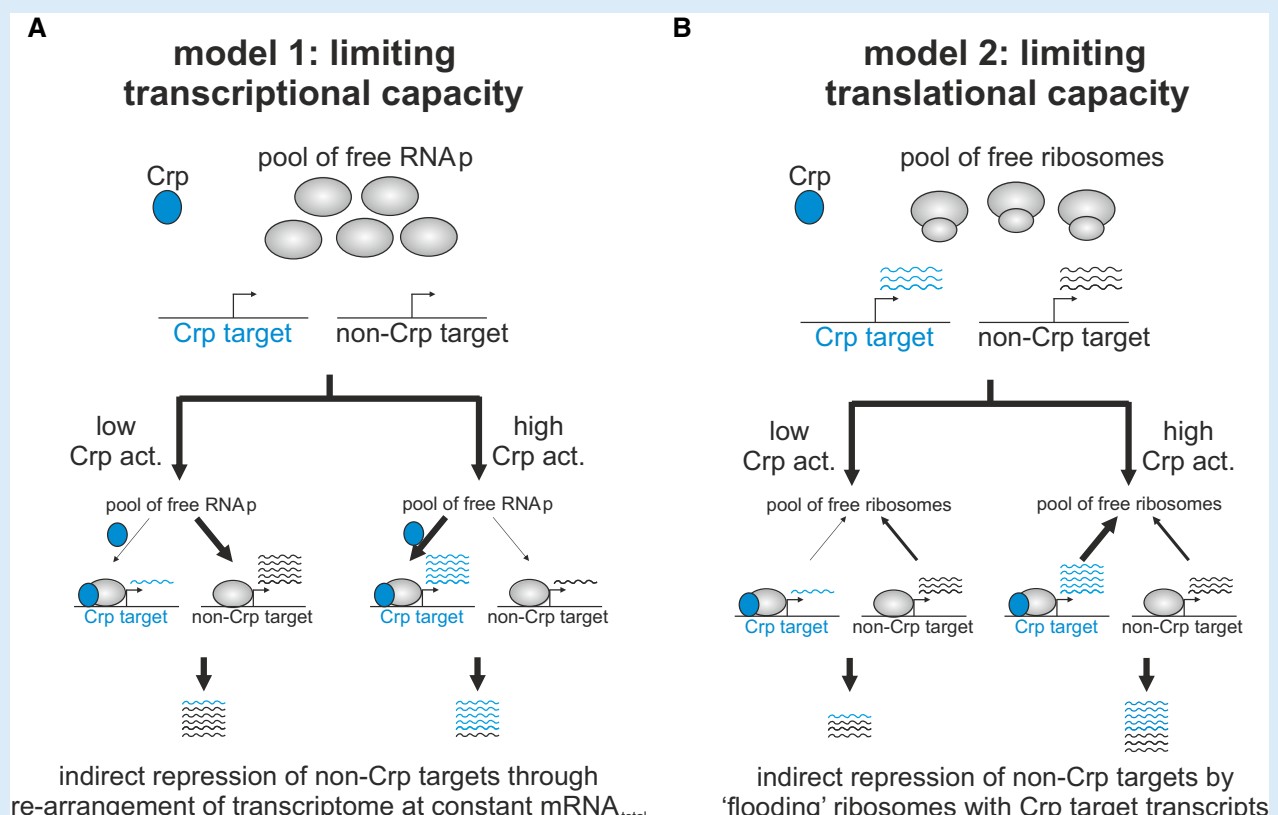

physiological rationale for the increased protein concentration of anabolic proteins despite a decrease in the net flux: Increasing the concentration of the catalyzing enzyme is an effective way to increase the net flux in a thermodynamically choked reaction involving high exchange flux (Noor *et al*, 2014). Therefore, by increasing the expression of transaminases in the anabolic limitation, cells can counter the reduction in driving force resulting from 2-oxoglutarate accumulation. Future studies, with broader metabolome coverage, may explore the importance of such changes in thermodynamic driving force in more detail.

Overall, the findings reported in this study point to a common theme, in which *E. coli* largely relies on simple heuristics, or "rules of thumb" to cope with environmental and genetic changes (Tversky & Kahneman, 1974; Towbin *et al*, 2017). Instead of specifically adjusting its gene expression program to meet exactly the imposed metabolic demands, the coordination of metabolic activity in *E. coli* is achieved by an approximate global gene regulatory response, which sets the system roughly in the desired direction. This response is further adjusted locally through changes in enzyme saturation. However, the downside of such a simple regulatory program is that it effectively makes each reaction involved more expensive (since enzymes are being expressed at higher levels than needed strictly to maintain the desired metabolic flux), with a potential cost to the steady-state growth rate (Scott *et al*, 2010; Hui *et al*, 2015; O'Brien *et al*, 2016).

While the approximate (or suboptimal) regulation of protein expression is widespread in bacteria (Price *et al*, 2013; Towbin *et al*, 2017), it appears to be at odds with numerous laboratory-based evolutionary studies (e.g., LaCroix *et al*, 2015), showing that even small differences in fitness (i.e., growth rate) are selected against. One previously proposed explanation (Price *et al*, 2013) is that cells have not evolved to cope with the artificial laboratory environments they are subjected to, which is particularly true for the genetically implemented limitations used in this study. Nevertheless, a recent study, which quantified the global proteome allocation of wild-type *E. coli* strains growing on various carbon sources (resulting in different degrees of catabolic limitation), found an inverse relationship between catabolic and anabolic proteome sectors that is consistent with our findings (Schmidt *et al*, 2016). Future studies may examine whether natural anabolic limitations similarly exert an effect on *E. coli*'s proteome allocation that is consistent with our genetically implement anabolic limitation.

A more parsimonious explanation of the difference between the degree of optimization manifested by the laboratory-evolved and natural strains is that the simple regulatory programs emerged from an evolutionary trade-off: Given the multitude of environments that cells could encounter, it is simply infeasible to have a dedicated optimized program for each environment that is also consistent across conditions (Shoval *et al*, 2012; Keren *et al*, 2013; Price *et al*, 2013). Consequently, cells may rather rely on using the "rule of thumb" as a heuristic guide to coarsely allocate the proteome according to a few signals (Chubukov *et al*, 2014). A good illustration of this strategy is the proteome response to anabolic limitation: If cells cannot identify the exact reaction responsible for the reduction in anabolic capacity (in this case, it is ultimately the reduced glutamate synthesis that slows down transamination reactions), or if it cannot fix the problem (the "optimal solution" would be to upregulate just the transaminases), the best alternative may be an across-the-board increase in the expression of anabolic proteins (which include all the transaminases). Importantly, proteome allocation within this mode of response can be easily coordinated and optimized (You *et al*, 2013; Hui *et al*, 2015). Future research will reveal the prevalence and nature of this type of simple regulatory strategies underlying microbial responses to complex environmental challenges.

# Materials and Methods

### Reagents and strains

Unless stated otherwise, all reagents were obtained from Sigma-Aldrich. All used strains were derived from NCM3722 (Soupene *et al*, 2003), a full list of strains is provided in Appendix Table S1, and a detailed description of the strains is provided in Appendix Text 1 and Appendix Fig S1. Fluorescent transcriptional reporter plasmids were obtained from Gerosa *et al* (2013, 2015) and Zaslaver *et al* (2006) and transferred to NQ1399 by electroporation as described previously (Kochanowski *et al*, 2013b).

### Cultivation

All experiments were performed using M9 minimal medium (Kochanowski *et al*, 2013b), supplemented with 2 g/l glucose. Cultivations were performed as follows: M9 medium batch cultures in 96-deep-well format plates (Kuehner AG, Birsfeld, Switzerland), containing the same inducer concentration as for the subsequent main culture (with the exception of slow-growing NQ393 with 10/20 μM IPTG in the main culture, which were cultivated with 30 μM IPTG in the preculture to avoid the emergence of mutations), were inoculated 1:50 from LB precultures and incubated overnight at 37°C under shaking. Subsequently, 96-deep-well plate cultures were inoculated with overnight cultures to a starting OD600 of 0.03–0.05 (total fill volume per well: 1.2 ml) and incubated at 37°C under shaking. Culture OD600s were monitored by OD600 sampling from parallel wells on the same deep-well plate and subsequent OD600 measurements using a plate reader (TECAN infinite M200, Tecan Group Ltd, Männedorf, Switzerland). Cultivation of strains bearing transcriptional reporter plasmids and calculation of promoter activity as the OD normalized GFP production rate were performed as described previously (Gerosa *et al*, 2013; Kochanowski *et al*, 2013b). Main cultures in M9 minimal medium with 2 g/l glucose with varying external cyclic AMP concentrations (ranging from 0 to 10 mM) were inoculated 1:100 with overnight cultures growing in M9 minimal medium with 2 g/l glucose and 1 mM cyclic AMP, and steady-state promoter activities were determined during the 1.5 h window during which the cultures exhibited the maximal growth rate. Steady-state GFP concentrations were calculated from promoter activities by division by the corresponding steady-state growth rate as described previously (Gerosa *et al*, 2013).

### Proteome analysis of NQ1399

Cultures of NQ1399 were grown in M9 medium with 2 g/l glucose at various cyclic AMP concentrations between 0 and 3 mM as described above. $^{15}$N reference cultures of NQ1399 were grown in modified M9 media (42.2 mM $Na_2HPO_4$, 22 mM $KH_2PO_4$, 8.56 mM $Na_2SO_4$,

11.3 mM $^{15}NH_4Cl$) at 0.01 and 3 mM cAMP. The samples were processed as described previously (Hui *et al*, 2015). The comparison of NQ1399 and wild type was performed as follows: Two replicate cultures were grown in $^{15}N$ reference M9 medium (in the presence of 0.2 mM cyclic AMP to match the wild-type growth rate). Then, protein samples of the 15N replicate cultures were mixed (50 μg per replicate) with 100 μg of an unlabeled wild-type protein sample from a matched condition (without the addition of cyclic AMP) and processed as described previously (Hui *et al*, 2015).

## Quantification of intracellular metabolite concentrations by untargeted metabolomics

Intracellular metabolomics samples were taken during mid-exponential phase at OD600s between 0.5 and 0.6 by fast filtration (sampling volume: 1 ml) (Link *et al*, 2013) and were immediately quenched in 4 ml quenching/extraction solution (40% methanol, 40% acetonitrile, 20% $H_2O$) at −20°C (Link *et al*, 2012). Samples were incubated for 2 h at −20°C, subsequently dried completely at 120 μbar (Christ RVC 2-33 CD centrifuge and Christ Alpha 2–4 CD freeze dryer), and stored at −80°C until measurements. Before measurements, samples were resuspended in 100 μl water, centrifuged for 5 min (5,000 *g*, 4°C) to remove residual particles, diluted 1:10 in water, and transferred to V-bottom 96-well plates (Thermo Fisher Scientific). Samples were measured by flow-injection time-of-flight mass spectrometry with an Agilent 6550 QToF instrument operated in negative ionization mode at 4 GHz high resolution in a range of 50–1,000 *m/z* as described before (Sévin & Sauer, 2014). Sample processing and ion annotation was performed based on accurate mass within 0.001 Da using the KEGG *E. coli* database (Ogata *et al,* 1999) as reference and accounting for single deprotonated forms of the respective metabolite (M–H$^+$) as described before (Fuhrer *et al*, 2011). Intensities of annotated ions were normalized to NCM3722 wild type to yield relative concentrations.

## Quantification of intracellular metabolite concentrations by targeted metabolomics

Intracellular metabolomics samples were taken as described above, with one difference: Immediately after quenching, 100 μl of a fully $^{13}C$-labeled *E. coli* internal metabolome extract was added for internal normalization. Samples were incubated and dried as described above. Before measurements, samples were resuspended in 100 μl water, centrifuged for 5 min (5,000 *g*, 4°C) to remove residual particles, and transferred to V-bottom 96-well plates (Thermo Fisher Scientific). Measurements, data acquisition, peak integration, and quantification of absolute metabolite concentrations were performed as described previously (Buescher *et al*, 2010; Kochanowski *et al*, 2017). To convert OD600 to cell volume, a conversion factor of 2.7 μl cell volume per mg CDW (Winkler & Wilson, 1966) (and a OD600 conversion factor to cell dry mass of 1 OD = 0.413 mg CDW/ml) was used.

## Quantification of uptake and secretion rates

Culture samples were taken at 6–8 time points together with parallel OD600 samples throughout exponential growth phase (sampling volume: 100 μl). Supernatants were separated from cells by centrifugation (5,000 *g*, 4 min, at 4°C) and transferred to V-bottom 96-well

plates (Thermo Fisher Scientific). Glucose and acetate concentrations in supernatants were determined by colorimetric enzymatic assays (Megazyme). All other secreted metabolites were quantified by flow-injection time-of-flight mass spectrometry as described above. Briefly, supernatants were diluted 1:10 in water and measured with an Agilent 6550 QToF instrument operated in negative ionization mode at 4 GHz high resolution in a range of 50–1,000 *m/z*. Sample processing and ion annotation was performed based on accurate mass within 0.001 Da using the KEGG *E. coli* database (Ogata *et al*, 1999) as reference and accounting for single deprotonated forms of the respective metabolite (M–H$^+$) as described before (Fuhrer *et al*, 2011). Absolute extracellular metabolite concentrations were determined using parallel dilution series of the respective metabolite in the same medium as calibration curves. Uptake and secretion rates were determined from extracellular metabolite concentrations, corresponding OD600 (conversion factor to cell dry mass: 1 OD = 0.413 g CDW/l), and corresponding growth rates by linear regression as described previously (Haverkorn van Rijsewijk *et al*, 2011). With the exception of 2-oxoglutarate, all other quantified metabolites were secreted in minute amounts (< 70 μmol/gCDW/h).

## Quantification of intracellular metabolic fluxes

Intracellular central metabolic fluxes were determined by $^{13}C$ flux analysis as follows. Cultivation was performed as described above, and glucose was added as the [1-$^{13}C$] isotope (> 99%; Cambridge Isotope Laboratories), or as a mixture of 20% (wt/wt) [U-$^{13}C$] (> 99%; Cambridge Isotope Laboratories) and 80% [$^{12}C$] isotopes. Labeling samples (sampling volume 1 ml) were taken during mid-exponential phase (OD600 0.5–0.7), cells were harvested by centrifugation (13,000 *g*, 3 min), and cell pellets were washed once in cold 0.9% NaCl and stored dry at −20°C. $^{13}C$ flux ratios were determined as described previously (Fischer, 2004; Zamboni & Fendt, 2009; Haverkorn van Rijsewijk *et al*, 2011). Briefly, cell pellets were hydrolyzed, dried, and derivatized, and labeling patterns of derivatized proteinogenic amino acids were quantified by GC-MS using a 6890 GC system combined with a 5973 Inert SL MS system (Agilent Technologies, Santa Clara, USA). Metabolic flux ratios were determined based on these labeling patterns (after correcting for naturally occurring $^{12}C$ as described in van Winden *et al* (2002)) using the software FiatFlux (Zamboni *et al*, 2005). Two flux ratios (glyoxylate shunt and malic enzyme flux) were found to be zero in all conditions and were discarded in subsequent analyses. Using the flux ratios, uptake/secretion rates, and the measured growth rate as inputs, absolute central metabolic fluxes were inferred using the software FiatFlux (Zamboni *et al*, 2005).

To infer metabolic fluxes beyond central carbon metabolism, flux balance analysis (FBA) was performed with the Cobra toolbox v3.0 (Schellenberger *et al*, 2011; Heirendt *et al*, 2019) in MATLAB (Version 2019A) using the *E. coli* genome-scale metabolic model iJO1366 (Orth *et al*, 2011). This model was further modified as follows: (i) Based on the measured metabolite exchange rates, a phenylpyruvate exchange reaction was added using the Cobra toolbox command addExchangeRxn. (ii) The reaction boundaries of glyoxylate shunt and malic enzyme fluxes were set to zero based on the $^{13}C$ labeling data. (iii) The reaction boundaries of two additional reactions ("FBA3" and "F6PA"), which emerged empirically as glycolytic bypass reactions that were not supported by the $^{13}C$

labeling data, were also set to zero. (iv) For those conditions in which the strain NQ393 was used (i.e., anabolic limitation), the reaction boundaries of the deleted enzyme glutamate dehydrogenase ("GLUDy") were set to zero. (v) All measured major (glucose, acetate, 2-oxoglutarate) and eight minor (succinate, fumarate, malate, citrate, glycine, valine, glutamate, and phenylpyruvate) exchange rates (with maximal exchange rates across conditions > 10 micromol/h/gCDW), as well as the measured growth rate and the oxygen consumption rate as inferred by $^{13}$C flux analysis (see above), were used as constraints (allowing for 5% deviation for major exchange rates, oxygen consumption rate and growth rate, as well as 10% deviation for minor exchange rates) (Mo *et al,* 2009). To enable fumarate secretion, the main fumarate transporter DctA was set to be reversible. The exchange rates of pyruvate, lactate, and ethanol (common secretion products that were not found to be secreted in any of the conditions) were constrained to zero. (vi) Four ratios of absolute fluxes (as determined above, namely phosphofructokinase <> G6P dehydrogenase; phosphoglucoisomerase <> phosphogluconate dehydratase; malate dehydrogenase <> PEP carboxylase; and enolase <> PEP carboxykinase) were incorporated as constraints using the Cobra toolbox command addRatioReaction. Using this modified and constrained model, FBA was performed in two steps. First, the ATP production rate was used as an objective function (Schuetz *et al,* 2007) and maximized using the Cobra toolbox command optimizeCBmodel (i.e., maximize flux of reaction "ATPM"). Second, using the obtained ATP production rate as an additional constraint, minimization of sum of fluxes was performed to yield the final inferred flux distribution using the Cobra toolbox command minimizeModelFlux. In both steps, Gurobi V9.0.2 (Gurobi Optimization) was used as the FBA solver, and the minNorm parameter was set to 1e-6. The quality of the FBA flux estimates was assessed empirically by comparison with $^{13}$C flux analysis data. For all non-zero fluxes, the uncertainty of the flux estimates in the constrained FBA model (including the ATP production rate obtained in the first FBA step as an additional constraint) was determined by flux variability analysis using the Cobra toolbox command fluxVariability (using the "fastSNP" option to prevent loops). Due to numerical issues, both the optimality percentage and the lower bound of the ATP production rate constraint had to be relaxed very slightly (from 100 to 99.99%).

### Regulation analysis

Regulation analysis was performed as described previously (Chubukov *et al,* 2013; Gerosa *et al,* 2015). Only reactions with non-zero fluxes in all tested conditions were considered. Absolute fluxes were normalized to NCM3722 wild type in the respective experiments and log-transformed. Transcriptional regulation coefficients were determined using a previously published proteomics study, which had determined relative protein concentrations in equivalent carbon and glutamate limitations (Hui *et al,* 2015). Relative protein concentrations were linearly interpolated to match exactly the same growth rate as in the corresponding flux measurements, normalized to NCM3722 wild type and log-transformed. Only proteins that were quantified in at least one limitation were considered. For each considered reaction, protein regulation coefficients were estimated separately for each limitation (by linear regression) as the slope between log-normalized fluxes and log-normalized protein concentrations. Reaction-protein pairs were used as defined in the aforementioned genome-scale metabolic model (Orth *et al,* 2011). For reactions that are associated to more than one isoenzyme, the final protein regulation coefficients were calculated as the average regulation coefficient across all measured proteins.

Enzyme saturation regulation coefficients were determined as follows. To account for potential non-linearity of the relationship between substrate concentration and flux (e.g., due to non-Michaelis–Menten type enzyme kinetics), approximate kinetic orders ($\alpha$) for substrates were estimated using the following equation across all conditions as described previously (Chubukov *et al,* 2013; Gerosa *et al,* 2015) (eq. 1):

$$\min_{0 \leq \alpha \leq 4} \log(J_i) - \log(P_i) = \sum_{x \in S_i} \alpha_{ix} \cdot \log(M_x)$$

where J denotes the normalized flux, P denotes the normalized protein concentration, and M denotes the reaction substrate(s) with corresponding kinetic order(s) $\alpha$. $\alpha$ was constrained to be between 0 and 4 to set a biologically realistic upper bound on the non-linear gain. Highly connected reactants (i.e., $H_2O$, $H^+$, $CO_2$, $HCO_3^-$, sulfate, phosphate, ammonia) were excluded from the analysis. Kinetic orders were estimated independently for each flux–enzyme pair (considering both limitations) by least square optimization using the *lsqlin* function of MATLAB. Only reactions for which all substrates had been quantified were considered. As above, for reactions that are connected to more than one isoenzyme, the final regulation coefficients were calculated as the average regulation coefficients across all measured proteins.

## Data availability

The raw cAMP titration proteomics data are available at proteomeXchange (accession number PXD024504) under the follow link: http://proteomecentral.proteomexchange.org/cgi/GetDataset?ID=PXD024504

The code necessary to reproduce the FBA, FVA, and regulation analysis results is available on GitHub under the following link: https://github.com/karl-kochanowski/MSB10064-Ecoli-metabolic_analysis

**Expanded View** for this article is available online.

### Acknowledgements

We thank for helpful discussions with various colleagues: Elad Noor on metabolism, Hualin Shi for possible mechanisms of Crp-mediated indirect repression, and Uri Alon for the "rule of thumb" strategy of regulation. This work was supported by the NIH (R01GM109069) and the NSF (MCB 1818384) to TH, by the NIH (R01GM118850) to JRW, and by the SignalX project of the Swiss Initiative for Systems Biology (SystemsX.ch).

### Author contributions

Conceived and designed the study: KK, TH. Performed experiments and analyses: KK, HO, VP, TH. Supervised the study: JW, US, TH.

### Conflict of interest

The authors declare that they have no conflict of interest.

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
