## [Review Process File · Molecular Systems Biology]

Global coordination of metabolic pathways in *Escherichia coli* by active and passive regulation

Karl Kochanowski, Hiroyuki Okano, Vadim Patsalo, James R. Williamson, Uwe Sauer, and Terence Hwa

DOI: [10.15252/msb.202010064](https://doi.org/10.15252/msb.202010064)

Corresponding author(s): Terence Hwa (hwa@ucsd.edu)

Review Timeline:

Submission Date:	15th Oct 20
Editorial Decision:	16th Nov 20
Revision Received:	3rd Feb 21
Editorial Decision:	10th Feb 21
Revision Received:	4th Mar 21
Accepted:	9th Mar 21

Editor: Maria Polychronidou

Transaction Report:

Thank you again for submitting your work to Molecular Systems Biology. We have now heard back from the two referees who agreed to evaluate your study. Overall, the reviewers are supportive. However, they raise a series of concerns, which we would ask you to address in a revision.

I think that the recommendations of the referees are rather clear and straightforward to address. Therefore, I see no need to repeat any of the points listed below. Please let me know in case you would like to discuss any of the issues raised. All issues raised by the referees would need to be satisfactorily addressed.

On a more editorial level, we would ask you to address the following points.

REFeree REPORTS

Reviewer #1:

The authors study the coordinated regulation of bacterial metabolic fluxes in response to nutrient limitation, using *E. coli* as a well suited model system. The phenomenology of the coordinated metabolic response to carbon and nitrogen limitation has largely been established, but the underlying mechanisms have been elusive. Aiming to fill in this important gap, the authors combined state-of-the-art mass spectrometry of metabolites and proteins with fluorescent reporter assays and metabolic flux analysis. To obtain full external control of both carbon and nitrogen limitation, the authors constructed strains that permit them to externally titrate carbon uptake and amino acid synthesis. These strains appear to recapitulate the known behavior of the wild-type system under carbon and nitrogen limitation. Overall, the research is professionally executed by leading experts in the field. The central conclusion is that the coordinated response in the metabolic fluxes is largely the combined result of global transcription regulation by Crp and "local adjustment" of the fluxes via passive changes in metabolite concentrations. These two contributions are found to rationalize about 50% of the observed metabolic flux changes. The authors adequately discuss both the

importance and the limitations of their findings. Therefore, there is no doubt in my mind that this manuscript deserves to be published in MSB. I do have a couple of questions and comments to the authors though:

(1) Regarding the massive secretion of 2-oxoglutarate under the genetically induced nitrogen limitation: It was not clear to me whether this was potentially an artefact of the titration of GOGAT to emulate anabolic limitation. The authors mentioned that this led to a reduction in biomass yield consistent with that observed in nitrogen-limited chemostat cultures. This statement appears to imply that 2-oxoglutarate secretion is not an artefact and will equally occur in the wild-type system, but simply has gone unnoticed so far. Is this implication intended? If yes, what are the further implications of this observation? For instance, in the case of acetate secretion, cells later switch to take up and utilize acetate again (after glucose is depleted) - is 2-oxoglutarate also reused later?

(2) To me, the flux-balance analysis applied to obtain the metabolic fluxes is largely a black box, which leaves me wondering what the level of confidence in the resulting fluxes should be. Could the authors convey some sense of the level of uncertainty in these fluxes, especially for the reader who has no first-hand experience with flux-balance analysis? For instance, in the end roughly 50% of the measured flux changes are not fully explained by the changes in protein levels and metabolite concentrations. How should one think about those 50%? If one takes all the metabolic fluxes inferred from flux-balance analysis very seriously, then there is major additional regulation going on, which future work will have to clarify. Or are the observations well within the margin of error, such that it is not clear that significant additional regulation must be happening?

(3) Given the (plausible) finding that Crp indirectly represses all non-Crp targets by diverting expression capacity to the Crp targets, it is actually interesting (from a mechanistic perspective) that a significant number of genes do not appear to change their expression much with growth rate under either carbon or nitrogen limitation (Hui et al, Mol Syst Biol, 2015). Does this mean that all of these genes must have an additional layer of regulation which enables them to maintain their expression level essentially constant? Is there perhaps a similarly generic and simple mechanism to achieve this?

(4) Regarding the regulation via changes of metabolite concentrations and level of saturation of the enzymes: I am still in a somewhat confused state of mind, even after reading the discussion section and the supplementary information sections 3 and 4. How can passive changes of the metabolite levels always automatically produce the intended changes in the fluxes? I understand that there is a nonlinear feedback effect and that this is passive, i.e. it does not require any additional regulation mechanism. But how come this passive feedback always goes in the right direction? (and amplitude?) Also, this requires that none of these metabolic enzymes is fully saturated in vivo. The definition of ρ_S in eq. S7, which relies on eq. S2, actually assumes non-saturated enzymes to begin with. Is this assumption consistent? In other words, does this assumption hold for those enzymes where the authors find a significant regulatory contribution from changes in metabolite level?

Reviewer #2:

The manuscript by Kochanowski and colleagues describes a comprehensive study on the proteomic and metabolomic responses to catabolic and anabolic limitations in *E. coli*. By titrating the level of cyclic AMP, they found that the catabolic repressor (CRP) not only directly regulate

catabolic enzymes as it was known for, but it also indirectly affects the expression of anabolic enzymes, presumably by shifting global resources for transcription or translation. The latter indirect effect presents an interesting 'mechanism' for reprogramming gene expression without direct regulatory connections, which is reminiscent of the effects of (p)ppGpp on constitutive promoters that the authors have previously demonstrated.

The authors further showed that the proteomic changes under nutrient limitation is insufficient to explain the changes in metabolic fluxes. For many enzymes, there are excess capacities at reduced growth rates, and the changes in substrate concentration are correlated to the flux changes. These results suggest that most enzymes are not operating near saturation, and that cells overproduce these proteins under nutrient limitation. This suboptimal proteome allocation may be a side effect of having only a coarse regulator (CRP) that controls both anabolic and catabolic enzyme production.

Understanding how cells allocate their proteome is one of the core questions in systems biology. The first half of this work provides a basis for how global reprogramming can be achieved by a single regulator through its direct and indirect effects. The second half of this work describes a simple principle of 'rules of thumb' for coping with environmental changes. Both observations are highly informative and will influence future studies on this subject. I support publication at Molecular Systems Biology with minor revision. Below are several minor suggestions.

1. This study achieves nutrient limitation by artificially titrating the expression of glucose transporter or glutamate synthase. While elegant, this approach raises a question of whether the proteomic response, and especially the overproduction of enzymes, is related to the inability to launch coordinated expression of catabolic or anabolic enzymes in these strains. In other words, is it possible that if the native regulation were intact, the growth would not be limited by a single bottleneck, but by a multitude of enzymes that are regulated together? It may be difficult to answer this question experimentally. Perhaps the authors could look into existing proteomic data on wildtype strains and see if there are situations in which the relative expression between the titrated enzyme and the non-saturated enzymes is as low as what was seen in this work.

2. The strain NQ1399 has noticeable upregulation in flagella and chemotaxis genes (Fig. S7B). Although it is unlikely to change the conclusions drawn from this strain, if the authors want to figure out the source of this upregulation, they may check if there are additional genetic differences in the promoter of *flhDC* or in *IrhA*, which are frequently mutated during strain construction (<https://doi.org/10.1128/JB.00259-19>).

Response to specific reviewer comments:**Reviewer #1:**

The authors study the coordinated regulation of bacterial metabolic fluxes in response to nutrient limitation, using *E. coli* as a well suited model system. The phenomenology of the coordinated metabolic response to carbon and nitrogen limitation has largely been established, but the underlying mechanisms have been elusive. Aiming to fill in this important gap, the authors combined state-of-the-art mass spectrometry of metabolites and proteins with fluorescent reporter assays and metabolic flux analysis. To obtain full external control of both carbon and nitrogen limitation, the authors constructed strains that permit them to externally titrate carbon uptake and amino acid synthesis. These strains appear to recapitulate the known behavior of the wild-type system under carbon and nitrogen limitation. Overall, the research is professionally executed by leading experts in the field. The central conclusion is that the coordinated response in the metabolic fluxes is largely the combined result of global transcription regulation by Crp and "local adjustment" of the fluxes via passive changes in metabolite concentrations. These two contributions are found to rationalize about 50% of the observed metabolic flux changes. The authors adequately discuss both the importance and the limitations of their findings. Therefore, there is no doubt in my mind that this manuscript deserves to be published in MSB. I do have a couple of questions and comments to the authors though:

Author response:

We thank this reviewer for their positive assessment of our work.

(1) Regarding the massive secretion of 2-oxoglutarate under the genetically induced nitrogen limitation: It was not clear to me whether this was potentially an artefact of the titration of GOGAT to emulate anabolic limitation. The authors mentioned that this lead to a reduction in biomass yield consistent with that observed in nitrogen-limited chemostat cultures. This statement appears to imply that 2-oxoglutarate secretion is not an artefact and will equally occur in the wild-type system, but simply has gone unnoticed so far. Is this implication intended? If yes, what are the further implications of this observation? For instance, in the case of acetate secretion, cells later switch to take up and utilize acetate again (after glucose is depleted) - is 2-oxoglutarate also reused later?

Author response:

This reviewer raises an interesting point. Given the consistent drop in biomass yield and 2-oxoglutarate accumulation between our GOGAT titration and N-limited chemostats, it is certainly conceivable that 2-oxoglutarate may also be secreted by N-limited chemostats. Although to our knowledge 2-oxoglutarate secretion has never been reported in N-limited chemostats, recent studies have already demonstrated that many metabolites other than acetate can be secreted by *E. coli* (e.g. PMID 22963408, PMID 23903661), notably including 2-oxoglutarate (PMID 22963408). Similarly, it is conceivable that *E. coli* may utilize secreted 2-oxoglutarate (according to literature, *E. coli* can grow on 2-oxoglutarate as the sole carbon source). However, we believe that testing these conjectures experimentally is beyond the scope of this manuscript.

In the revised manuscript, we now discuss the 2-oxoglutarate secretion data in more detail (lines 188-191).

(2) To me, the flux-balance analysis applied to obtain the metabolic fluxes is largely a black box, which leaves me wondering what the level of confidence in the resulting fluxes should be. Could the authors convey some sense of the level of uncertainty in these fluxes, especially for the reader who

has no first-hand experience with flux-balance analysis? For instance, in the end roughly 50% of the measured flux changes are not fully explained by the changes in protein levels and metabolite concentrations. How should one think about those 50%? If one takes all the metabolic fluxes inferred from flux-balance analysis very seriously, then there is major additional regulation going on, which future work will have to clarify. Or are the observations well within the margin of error, such that it is not clear that significant additional regulation must be happening?

Author response:

This reviewer raises an interesting question. In order to assess the uncertainty of our flux estimates, we adapted our flux balance analysis pipeline such that it also enables flux variability analysis (FVA). FVA allows to determine for each flux the range of values that are compatible with the constraints set by the physiology data, and thus provides a measure of the flux uncertainty. Our FVA analysis revealed that for the vast majority of FBA estimates, the flux uncertainty is indeed minimal, consistent with previous work in yeast (PMID 27789812).

In the revised manuscript, we now include the FVA results in Appendix Figure S13D. Moreover, we greatly expanded our description of the FBA and FVA methods to clarify this aspect of our work. In addition, we included the source code necessary to reproduce the FBA and FVA results in the revision documents, and will make it publicly available in GitHub prior to publication.

(3) Given the (plausible) finding that Crp indirectly represses all non-Crp targets by diverting expression capacity to the Crp targets, it is actually interesting (from a mechanistic perspective) that a significant number of genes do not appear to change their expression much with growth rate under either carbon or nitrogen limitation (Hui et al, Mol Syst Biol, 2015). Does this mean that all of these genes must have an additional layer of regulation which enables them to maintain their expression level essentially constant? Is there perhaps a similarly generic and simple mechanism to achieve this?

Author response:

This reviewer is right, our results do suggest that proteome sectors which deviate from the expression patterns of anabolic proteins (and are not Crp-regulated) have additional regulatory layers. In the manuscript, we discuss one such proteome sector (ribosomal proteins), which strictly correlate with the growth rate in both catabolic and anabolic limitations, and for which we already propose a putative mechanism (i.e. regulation by ppGpp). For other expression patterns, in particular the “constitutive”-like pattern this reviewer is referring to, we currently do not know the underlying mechanism.

In the revised manuscript, we expanded our discussion of this point (lines 302-305).

(4) Regarding the regulation via changes of metabolite concentrations and level of saturation of the enzymes: I am still in a somewhat confused state of mind, even after reading the discussion section and the supplementary information sections 3 and 4. How can passive changes of the metabolite levels always automatically produce the intended changes in the fluxes? I understand that there is a nonlinear feedback effect and that this is passive, i.e. it does not require any additional regulation mechanism. But how come this passive feedback always goes in the right direction? (and amplitude?) Also, this requires that none of these metabolic enzymes is fully saturated in vivo. The definition of ρ_S in eq. S7, which relies on eq. S2, actually assumes non-saturated enzymes to begin with. Is this assumption consistent? In other words, does this assumption hold for those enzymes where the authors find a significant regulatory contribution from changes in metabolite level?

Author response:

We apologize for the lack of clarity in the original submission. The regulation analysis employed in this work is best understood as a systematic “consistency check” of flux/protein/metabolite changes

for individual reactions. This consistency check involves fits to simplified forms of the actual enzyme kinetics (in which a single parameter, α , encapsulates the impact of each substrate on a given reaction). These fits enable us to test whether – given the uncertainty of the actual in vivo enzyme kinetics – the discrepancy between flux and protein changes can be accounted for by the observed changes in substrate concentration. Using full mechanistic enzyme kinetic models would require knowledge of the in vivo kinetic parameters, which are mostly inaccessible.

This reviewer is right, such a “passive regulation” mechanism requires non-saturated enzymes: in this case, substrates effectively act as buffers, where at a given flux v a decrease in E will automatically lead to an increase in substrate concentration to maintain balanced flux. Our data alone does not allow to determine whether any given enzyme is non-saturated (since we only have relative metabolite concentrations available). Nevertheless, recent studies suggest that many metabolic enzymes in *E. coli* are non-saturated in vivo (e.g. PMID 27351952).

In the revised manuscript, we expanded the description of the approach in the appendix (section 3). In addition, we included the source code necessary to reproduce the regulation analysis results in the revision documents, and will make it publicly available in GitHub prior to publication.

Reviewer #2:

The manuscript by Kochanowski and colleagues describes a comprehensive study on the proteomic and metabolomic responses to catabolic and anabolic limitations in *E. coli*. By titrating the level of cyclic AMP, they found that the catabolic repressor (CRP) not only directly regulate catabolic enzymes as it was known for, but it also indirectly affects the expression of anabolic enzymes, presumably by shifting global resources for transcription or translation. The latter indirect effect presents an interesting ‘mechanism’ for reprogramming gene expression without direct regulatory connections, which is reminiscent of the effects of (p)ppGpp on constitutive promoters that the authors have previously demonstrated.

The authors further showed that the proteomic changes under nutrient limitation is insufficient to explain the changes in metabolic fluxes. For many enzymes, there are excess capacities at reduced growth rates, and the changes in substrate concentration are correlated to the flux changes. These results suggest that most enzymes are not operating near saturation, and that cells overproduce these proteins under nutrient limitation. This suboptimal proteome allocation may be a side effect of having only a coarse regulator (CRP) that controls both anabolic and catabolic enzyme production.

Understanding how cells allocate their proteome is one of the core questions in systems biology. The first half of this work provides a basis for how global reprogramming can be achieved by a single regulator through its direct and indirect effects. The second half of this work describes a simple principle of ‘rules of thumb’ for coping with environmental changes. Both observations are highly informative and will influence future studies on this subject. I support publication at Molecular Systems Biology with minor revision. Below are several minor suggestions.

Author response:

We thank this reviewer for their positive assessment of our work.

1. This study achieves nutrient limitation by artificially titrating the expression of glucose transporter or glutamate synthase. While elegant, this approach raises a question of whether the proteomic response, and especially the overproduction of enzymes, is related to the inability to launch coordinated expression of catabolic or anabolic enzymes in these strains. In other words, is it possible that if the native regulation were intact, the growth would not be limited by a single

bottleneck, but by a multitude of enzymes that are regulated together? It may be difficult to answer this question experimentally. Perhaps the authors could look into existing proteomic data on wildtype strains and see if there are situations in which the relative expression between the titrated enzyme and the non-saturated enzymes is as low as what was seen in this work.

Author response:

We agree with this reviewer that our usage of titration strains – albeit highly useful to generate gradual catabolic/anabolic limitations – does come with caveats. Recent proteomics efforts using wildtype *E. coli* strains (PMID 26641532) show at least for various carbon sources (= catabolic limitation) a similar inverse expression pattern for AA metabolism and e.g. respiratory proteins (which are part of C-sector). Unfortunately, this study did not include any anabolic limitations (i.e. nitrogen-limited chemostats), and we are also not aware of any other large-scale proteomics data set in *E. coli* which includes anabolic limitations. Therefore, the question of whether the inverse coordinated expression of catabolic and anabolic proteins extends to more natural anabolic limitations remains open.

In the revised manuscript, we expanded the discussion of our study's caveats to reflect this reviewer's concern (lines 348-353).

2. The strain NQ1399 has noticeable upregulation in flagella and chemotaxis genes (Fig. S7B). Although it is unlikely to change the conclusions drawn from this strain, if the authors want to figure out the source of this upregulation, they may check if there is additional genetic differences in the promoter of *flhDC* or in *IrhA*, which are frequently mutated during strain construction (<https://doi.org/10.1128/JB.00259-19>).

Author response:

This reviewer raises an interesting point. We did not sequence the strains, therefore it is certainly possible that some expression changes we observe are caused by acquired mutations. In this particular case, we note that we observed a **down**regulation in flagella/chemotaxis genes, which nevertheless may be caused by mutations in *flhDC* or in *IrhA*.

In the revised manuscript, we now include this point as well as reference suggested by this reviewer (lines 136-138).

1st Revision - Editorial Decision**10th Feb 2021**

Thank you again for sending us your revised manuscript. We think that the performed revisions satisfactorily address the reviewers' concerns and we are glad to inform you that we can soon accept the study for publication.

Before we can formally accept the manuscript we would ask you to address a few remaining editorial issues listed below.

2nd Authors' Response to Reviewers**4th Mar 2021**

The authors have made all requested editorial changes.

2nd Revision - Editorial Decision**9th Mar 2021**

Thank you again for sending us your revised manuscript. We are now satisfied with the modifications made and I am pleased to inform you that your paper has been accepted for publication.

Corresponding Author Name: Karl Kochanowski, Terence Hwa

Manuscript Number: MSB-2020-10064